fluid mechanics/nanotechnology

one-directional flow, carbon nanotube nanowire electrode, alternating current electric field, particular frequency

**Authors for correspondence:**
WooSeok Choi
e-mail: w.choi@ut.ac.kr
Geunbae Lim
e-mail: limmems@postech.ac.kr

# One-directional flow of ionic solutions along fine electrodes under an alternating current electric field

Jung Hwal Shin[1], Kanghyun Kim[2], Hyeonsu Woo[2], In Seok Kang[3], Hyun-Wook Kang[4], WooSeok Choi[5] and Geunbae Lim[2]

[1]School of Mechanical Engineering, Kyungnam University, 7 Kyungnamdaehak-ro, Masanhappo-gu, Changwon, Gyeongsangnam-do 51767, South Korea
[2]Department of Mechanical Engineering, and [3]Department of Chemical Engineering, Pohang University of Science and Technology (POSTECH), San31, Hyoja-dong, Pohang, Gyeongsangbuk-do 790-784, South Korea
[4]Biomedical Engineering, School of Life Sciences, Ulsan National Institute of Science and Technology (UNIST), 50, UNIST-gil, Ulsan 44919, South Korea
[5]Department of Mechanical Engineering, Korea National University of Transportation, 50 Daehak-Ro, Chungju, Chungcheongbuk-do 380-702, South Korea

 JHS, 0000-0002-8927-9356

Electric fields are widely used for controlling liquids in various research fields. To control a liquid, an alternating current (AC) electric field can offer unique advantages over a direct current (DC) electric field, such as fast and programmable flows and reduced side effects, namely the generation of gas bubbles. Here, we demonstrate one-directional flow along carbon nanotube nanowires under an AC electric field, with no additional equipment or frequency matching. This phenomenon has the following characteristics: First, the flow rates of the transported liquid were changed by altering the frequency showing Gaussian behaviour. Second, a particular frequency generated maximum liquid flow. Third, flow rates with an AC electric field (approximately nanolitre per minute) were much faster than those of a DC electric field (approximately picolitre per minute). Fourth, the flow rates could be controlled by changing the applied voltage, frequency, ion concentration of the solution and offset voltage. Our finding of microfluidic control using an AC electric field could provide a new method for controlling liquids in various research fields.

# 1. Introduction

Liquid control is important in a variety of research fields, including biological assays [1,2], drug delivery [3,4] and microchannel fluidics [5,6]. In general, closed systems are used to guide fluids, but such systems exhibit several problems including high flow resistance and frequent clogging. Recently, the phenomenon in which ionic solutions can flow along the outer surface of electrodes under a direct current (DC) electric field was reported [7,8].

In 1839, Faraday *et al*. were the first to transport liquid along an electrode surface; DC electric fields of several tens of kilovolts were employed [9]. Recently, Huang *et al*. reported that the applied voltage could be reduced to only several volts when the electrode diameter was of the order of hundreds of nanometres [8]. The flow was electro-osmotic in nature, induced by movement of an electric double layer near the electrode. The flow rates were of the order of picolitres per minute [7]. Given such flow rates afforded by DC liquid pumping, the technique is applicable to single cell studies, but not yet to studies on intact tissues. Therefore, liquid control technologies featuring larger flow rates are needed.

Generally, such a DC electric field can generate side effects above a certain voltage such as gas bubbles, electrode degradation/dissolution, hydrodynamic instability and sample contamination due to Faradaic electrochemical reactions near the electrodes [10]. However, these unwanted electrochemical reactions can be reduced with an alternating current (AC) electric field, which offers unique advantages, including the ability to drive fast programmable flows.

Here, we report a new phenomenon affording a wider range of flow rates: fluid motion under an AC electric field along a carbon nanotube (CNT) nanowire. To explore the motion, we varied the applied voltage and frequency, the concentration of KCl solution, and the offset voltage. The flow rates were analysed, and the frequency at which the flow rate was maximal was identified.

# 2. Experimental set-up

Figure 1 shows the experimental set-up, consisting of a humidity chamber, a CNT nanowire, a gold electrode, three-axis stages, light sources, optical microscopes and a function generator (AFG3101, Tektronix Inc., USA). The relative humidity was maintained above 85% to prevent evaporation of the liquid droplet on the gold electrode and the transported liquid along the CNT nanowire. To give the gold electrode surface hydrophobic characteristics, the gold electrode was coated with a 2% FPTE solution (Teflon AF 601S1-100-6, Dupont, USA) diluted in DC-75 (Acros Organics, Belgium) and cured in a 60°C oven for 10 min. A 2 µl liquid droplet was placed on the gold electrode and a CNT nanowire was slowly dipped into this droplet using a three-axis stage. In this study, a KCl ionic solution having similar mobility between $K^+$ and $Cl^-$ was used. And, the ionic concentration of the solution ranged from 0.01 to 100 mM. An AC electric field was applied between the CNT nanowire and the gold electrode by the function generator. The transported liquids along the CNT nanowires were observed an optical microscope, as indicated by the arrows in figure 1, and the flow rates were analysed by the volume change of the liquid droplet on the gold electrode.

# 3. Fabrication of carbon nanotube nanowires and liquid pumping experiments

In this study, single-walled carbon nanotubes (SWNTs) (1.0–1.2 nm diameter and 5–20 µm length) (Handhwa Nanotech, Seoul, South Korea) were prepared to make CNT nanowire electrodes (CNEs). SWNTs were dispersed by strong acid oxidization. First, SWNTs were put in a solution of sulfuric acid and nitric acid (3 : 1), and the solution was sonicated for 2–3 h. Second, the solution was mixed with a large amount of deionized (DI) water for neutralization. Third, the undispersed SWNTs were removed by vacuum filtration using a 0.22 µm filter (GVWP, Millipore). Finally, well-dispersed SWNTs were prepared by dispersing in pure DI water.

Cylindrical tungsten microwires (diameter: 300 µm) were sharpened in a NaOH electrolytic solution using a dynamic electrochemical etching method. The tungsten microwires were moved up and down slowly and repeatedly at DC 7 V to smooth their morphology. CNT nanowires were fabricated onto these sharpened tungsten microwires as follows. First, the pointed part of the tungsten microwire was immersed in a CNT-dispersed solution. Next, an AC electric field was applied between the tungsten microwire and the CNT solution, and the tungsten microwire was slowly withdrawn from the CNT

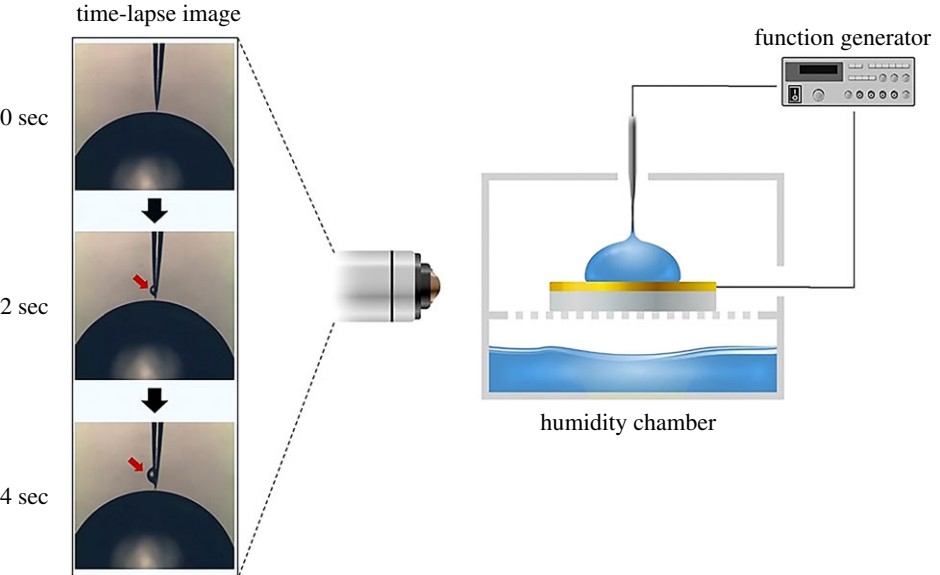

**Figure 1.** System set-up to observe the phenomenon. A carbon nanotube nanowire electrode (CNE) and a gold electrode were placed in a humidity chamber (greater than 85% relative humidity). When the AC electric field was applied between the two electrodes, one-directional flow was generated along the CNE. Time-lapse images show the transported liquid on the CNE. The generating droplet gradually increased over time.

solution. Through this step, CNTs gathered at the tip apex of the tungsten microwire by dielectrophoresis (DEP) force. As time passed, the solution was evaporated and the assembled CNTs were compressed by surface tension. The diameters of CNT nanowires could be controlled by the applied voltage, frequency and withdrawal velocity.

To improve the strength of CNT nanowires, two processes were performed. First, the CNT nanowires were fabricated in a laboratory environment. So, organic matter floating in the air was easily melted into a CNT dispersed solution. To remove this organic matter from the CNT nanowires, the general organic cleaning process including an acetone cleaning, methanol rinse and DI water rinse was conducted. Then, the solvent was removed by drying in a 60°C oven for 20 min. Second, gold nanoparticles were electrodeposited on the CNT nanowires. The solution containing gold nanoparticles was prepared by combining 5 mM $HAuCl_4 \cdot 4H_2O$ and 500 mM $HBO_3$. The amount and size of the generated Au nanoparticles onto the CNT nanowire could be controlled by the applied voltage and time. Au nanoparticles were evenly distributed onto the CNT nanowire surface at DC 1 V and 20 s condition. The coated Au nanoparticle size was about 10–100 nm [11]. The fabricated CNT nanowires coated by gold nanoparticles were about 800 nm in diameter.

A gold electrode was used as a substrate to place an ionic solution. Chromium (Cr, 10 nm) and gold (Au, 300 nm) were deposited onto a silicon substrate using thermal evaporation, and this substrate was used as a gold electrode.

In this study, we repeated the liquid pumping experiments seven times ($n = 7$) using different CNT nanowires. We varied the voltage, frequency, offset voltage and molar ionic concentration. The electronic supplementary material contains a movie of AC liquid pumping.

# 4. Results and discussion

## 4.1. Liquid pumping phenomenon under an alternating current condition

Figure 2 shows a schematic diagram of the transported liquid along a CNT nanowire under a DC electric field (a) and an AC electric field (b). In the DC condition, the flow direction can be controlled by changing the voltage bias [7]. The liquid flowed to the electrode having a negative bias. Liquid flow was generated above 1.5 V; a voltage lower than 3.5 V was used to avoid water electrolysis.

Generally, an AC electric field has a sine waveform; thus, the voltage bias changes periodically over time as a function of the frequency (figure 2b). In our studies, ionic solutions did not flow along a CNT nanowire under most AC conditions. However, one-directional flow was generated at a specific voltage

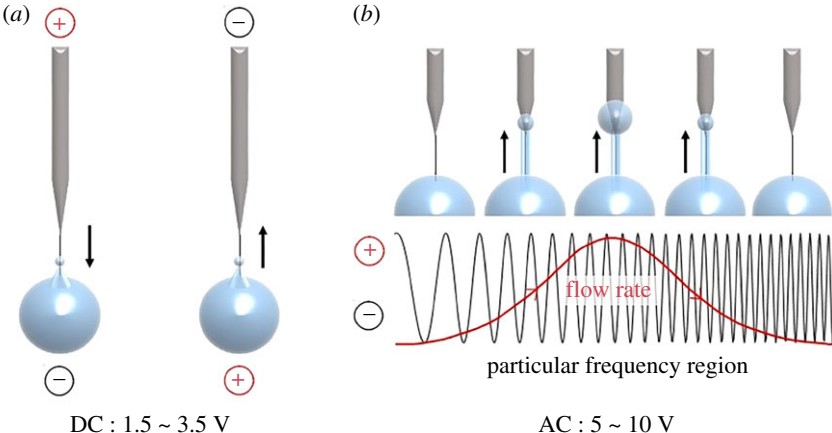

**Figure 2.** Schematic diagram of controlling liquid flow using direct current (DC) and alternating current (AC) electric fields. (*a*) DC conditions: liquid flowed to the electrode having a negative bias. (*b*) AC conditions: one-directional flow was generated above 5 V and in the MHz frequency range. The flow rate of the transported liquid was maximized at a particular frequency.

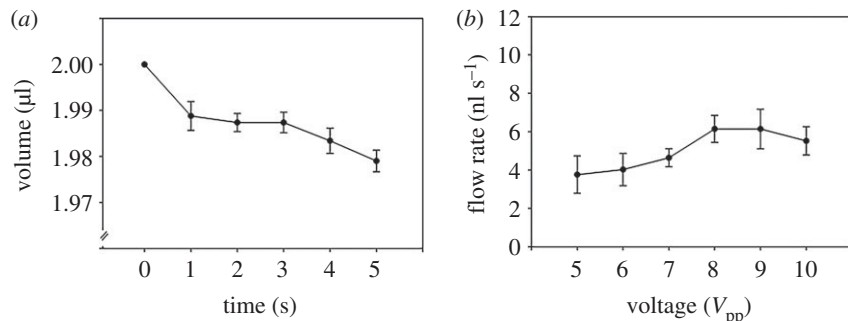

**Figure 3.** Flow rates with changing applied voltage. (*a*) Change in mother droplet volume over a 5 s time period (frequency: 60 MHz; ion concentration: 50 mM; and offset voltage: 0 V). (*b*) Flow rates as the applied voltage varied from 5 to 10 $V_{pp}$. The flow rates increased with the applied voltage from 5 to 8 $V_{pp}$ and were proportional to the power ($R^2 = 0.99$). Values were analysed by paired *t*-test, mean $\pm$ standard error, $n = 7$, *$p < 0.05$.

and frequency range. A peak-to-peak voltage of 5 $V_{PP}$ and a frequency of the order of tens of MHz initiated the liquid flow along the CNT nanowire. Importantly, the one-directional flow was generated when both conditions were satisfied.

Several interesting points were observed under the AC conditions as follows. First, the flow rates of the transported ionic solutions varied with the frequency; the variation showed Gaussian behaviour. Second, there was a particular frequency at which the flow rate was maximized. Third, the liquid flows of the AC conditions were much faster ($10^3$-fold) than that under DC conditions. The flow rates under AC and DC conditions were approximately nanolitre per minute and approximately picolitre per minute, respectively.

## 4.2. Flow rates of transported liquids when changing the applied voltage

Under DC conditions, a Faradaic reaction can be generated, as well as water electrolysis, above a certain voltage. Because water electrolysis produces $H_2$ and $O_2$ gas bubbles, the voltage range should be chosen carefully [7]. However, under AC conditions, such Faradaic reactions are negligible when the frequency is sufficiently high [12]. As mentioned before, one-directional flow was generated above 5 $V_{PP}$ and tens of MHz frequency. In these domains, the Faradaic reaction can be neglected, and we also confirmed experimentally that these conditions did not generate gas bubbles and pH change (electronic supplementary material, figure S1).

Figure 3 shows the experimental results when changing the applied voltage. The initial volume of KCl solution placed on the gold electrode was 2 μl. As time passed, liquid flowed along the CNT nanowire under the applied AC electric field. The transported liquid formed a thin 'precursor film'; several droplets were generated at random positions on the CNT nanowire. As the small droplets became bigger, they experienced more gravity force. When the downward force of gravity was larger

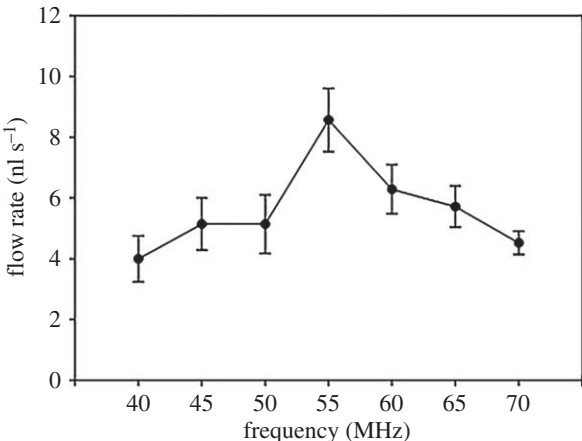

**Figure 4.** Flow rates when changing the applied frequency. Flow rates when changing the applied frequency from 40 to 70 MHz (voltage: 8 $V_{pp}$, ion concentration: 50 mM and offset voltage: 0 V). The flow rate was maximized at 55 MHz and significantly larger than that at 40 and 70 MHz. Values were analysed by a paired *t*-test, mean $\pm$ standard error, $n = 7$, $*p < 0.05$.

than the upward force caused by the electric field, the generated droplet flowed downward to the mother droplet. In this study, the time range from 0 to 5 s was adopted to calculate the flow rate of transported liquid during 1 min (observed time period) to exclude the gravity force effect. Figure 3*a* shows a mother droplet volume for 5 s. As time passed, the mother droplet volume decreased; most of the downward flow was initiated after 5 s.

Figure 3*b* shows the flow rates of the transported liquid along CNT nanowires when changing the applied voltage from 5 to 10 $V_{PP}$, and the other conditions remained fixed (frequency: 60 MHz, ion concentration: 50 mM, offset voltage: 0 V). Raw data about this experiment were added in electronic supplementary material, figure S2. Liquid transportation did not occur under conditions of less than 5 $V_{PP}$. The flow rates increased with increasing applied voltage from 5 to 8 $V_{PP}$, and the flow rate at the 8 $V_{PP}$ condition was significantly larger than that at 5 and 6 $V_{PP}$ ($p = 0.015$ and 0.009, respectively) conditions. The flow rate at 9 $V_{PP}$ was similar to that at 8 $V_{PP}$, and the flow rate at 10 $V_{PP}$ was slightly lower than that at 9 $V_{PP}$.

## 4.3. Flow rates of transported liquids when changing the applied frequency

Figure 4 shows the experimental results when changing the applied frequency. First, frequency ranges for generating liquid transportation were evaluated from 0 to 250 MHz. The applied voltage, ionic concentration of the KCl solution, and the offset voltage were 8 $V_{PP}$, 50 mM and 0 V, respectively. Raw data about this experiment were added in electronic supplementary material, figure S3.

Frequency ranges in the approximately hertz and approximately kilohertz domains did not generate any flows along CNT nanowires. The liquid flow was generated from 1 MHz (observed by optical microscope), but the liquid flow generated under 40 MHz was too small to change the volume of the mother droplet during 1 min. For this reason, the flow rate was calculated from 40 MHz. Other frequency domains of lower than 40 MHz and higher than 100 MHz did not cause any volume difference in the mother droplet. The flow rates increased from 40 to 55 MHz and decreased from 55 to 70 MHz. The flow rate at a particular frequency, 55 MHz, was larger than that of other conditions, and significantly larger than that of 40 and 70 MHz ($p = 0.006$ and 0.009, respectively). This tendency was similar to normal Gaussian curve.

## 4.4. Flow rates of transported liquids when changing the applied offset voltage

Figure 5 shows the experimental results for the flow rates as a function of the applied offset voltage; the other conditions remained fixed (voltage: 8 $V_{PP}$; frequency: 60 MHz; ion concentration: 50 mM). Raw data about this experiment were added in electronic supplementary material, figure S4. Liquid transportation along the CNT nanowire was not generated under conditions lower than $-1.0$ or higher than 1.0 V. The flow rates from $-0.5$ to 1.0 V were similar, but the flow rate at $-1.0$ V was

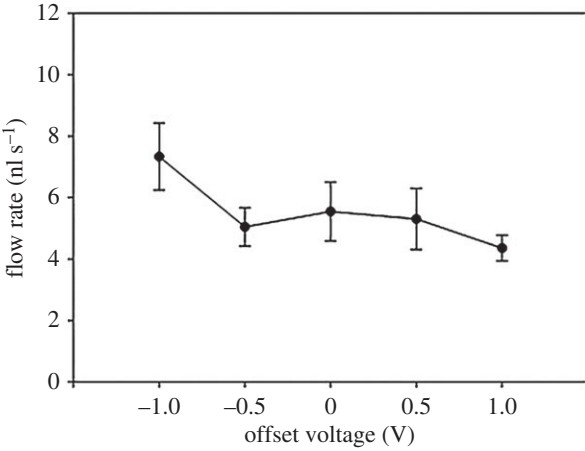

**Figure 5.** Flow rates as a function of the offset voltage. Flow rates as the offset voltage varied from −1.0 to 1.0 V (voltage: 8 $V_{pp}$; frequency: 60 MHz; and ion concentration: 50 mM). Values were analysed by a paired *t*-test, mean ± standard error, $n = 7$, *$p < 0.05$.

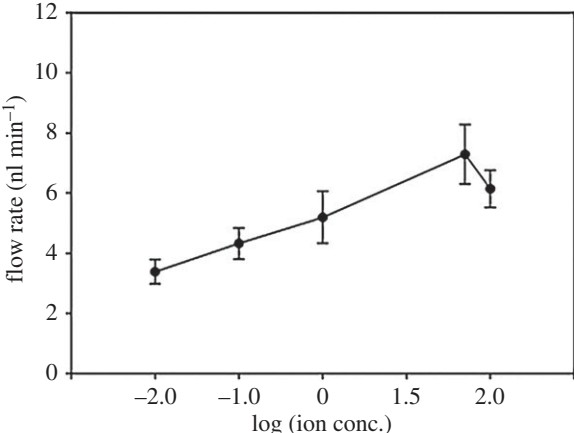

**Figure 6.** Flow rates as a function of the ion concentration. Flow rates as the ion concentration of the KCl solution varied from 0.01 to 100 mM (voltage: 8 $V_{pp}$; frequency: 60 MHz; and offset voltage: 0 V). Values were analysed by a paired *t*-test, mean ± standard error, $n = 7$, *$p < 0.05$.

different from that under other conditions. The flow rate at −1.0 V was the highest among −1.0 to 1.0 V conditions, significantly higher than that at 1.0 V ($p = 0.017$).

## 4.5. Flow rates of transported liquids when changing the ion concentration

Figure 6 shows the flow rates as a function of the ionic concentration of the KCl solution; the other conditions remained fixed (voltage: 8 $V_{pp}$; frequency: 60 MHz; offset voltage: 0 V). Raw data about this experiment were added in electronic supplementary material, figure S5. The ionic concentration of the KCl solution varied from 0.01 to 100 mM. Under all conditions, one-directional flow was generated along the CNT nanowire, and the flow rates of the transported liquid changed with the ion concentration. Flow rates increased with the ionic concentration of the KCl solution (0.01–50 mM); however, at 100 mM, the flow rate decreased slightly compared with that observed at 50 mM. The maximum flow rate under the 50 mM condition was significantly higher than that at 0.01 and 0.1 mM ($p = 0.01$ and 0.036, respectively).

## 4.6. Frequency ranges for generating one-directional flow changing molar concentrations

Figure 7 shows the frequency ranges generating liquid transportation for changes in the ionic concentration from 0.01 to 100 mM KCl solution (voltage: 5 $V_{pp}$; frequency 60 MHz; offset voltage: 0 V) with CNT nanowires. Raw data about this experiment were added in electronic supplementary material, figure S6. The frequency ranges for 50 and 100 mM were 64–173 and 57–185 MHz,

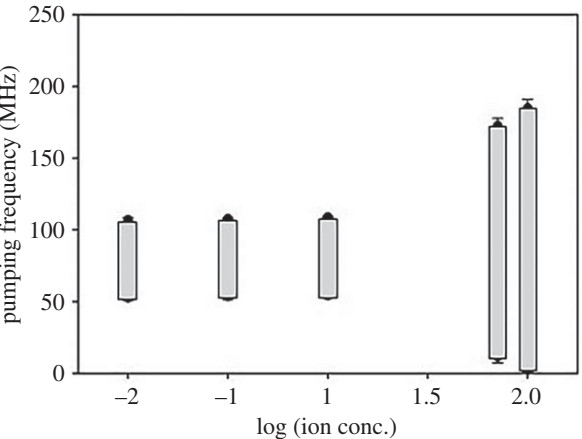

**Figure 7.** Pumping frequency ranges generating one-directional flow with CNEs (voltage: 5 V$_{pp}$; frequency: 60 MHz; and offset voltage: 0 V) (*n* = 5).

respectively, and the frequency ranges for 0.01, 0.1 and 1 mM were 87–107, 86–108 and 82–109 MHz, respectively. The frequency ranges for generating one-directional flow was reduced as the ionic concentration decreased from 1 to 100 mM. From 0.01 to 1 mM, the frequency ranges for generating liquid flow were similar. In the future, the method proposed in this manuscript could be applied to various bio-experiments such as drug injection.

## 4.7. Comparison with previous theories

Our experiment is very similar to a typical electrowetting experiment, in that a mother droplet is placed on the plane electrode and a needle electrode is placed on the opposite side [13]. Electrowetting is used to modify, move, mix and split droplets on a patterned electrode by adjusting the surface tension over a three-phase contact line. Several numerical analyses and theoretical studies have explored fluid movement via electrowetting under an AC electric field [14–16]. However, electrowetting differs from the phenomenon we describe in that an insulating layer is placed between the mother droplet and the plane electrode; this renders monodirectional flow impossible.

Here, the electric field is concentrated near the CNT electrode and the temperature gradient is generated via Joule heating. This gradient renders liquid permittivity, conductivity and mass density non-uniform, triggering an electrohydrodynamic flow, termed electrothermal flow, by exploiting the Koretweg–Helmholtz body force as follows:

$$f_{KH} = \rho_f E - \frac{1}{2}E^2\nabla\varepsilon + \nabla\left[\frac{1}{2}E^2\left(\frac{\partial\varepsilon}{\partial\rho}\right)\rho\right],$$

where $\rho_f$ is the free charge density, $\mathbf{E}$ is the electric field, $E = |\mathbf{E}|$ is the electric field strength, $\epsilon$ is the electrical permittivity and $\rho$ is the electrolyte mass density [17–19]. In this experiment, the temperature near the CNT electrode (where the electric field is concentrated) is higher than that inside the mother droplet. Therefore, flow from the CNT electrode to the plane electrode occurs within the droplet [16,20]. This flow is in the opposite direction to what we observed: flow developed along the electrode. Thus, it is difficult to explain the phenomenon by reference to electrothermal flow. Moreover, the electrothermal flow rate is proportional to the fourth power of the voltage, unlike what we observed experimentally.

Some studies, including Ramos *et al.*, reported an AC electro-osmosis phenomenon by which flow proceeds in one direction under an AC electric field because of induced charge electro-osmosis [21–29]. When an AC electric field is applied to a pair of electrodes, a diffuse double layer forms at the electrode surface if the frequency is sufficiently low to allow the ions to locally equilibrate. Induced charges in the double layer are subjected to an upward force by the tangential component of the electric field. Figure 8 shows the charge induced in response to applied potentials, the resulting electric field in the droplet, and the force on the induced charges. The force direction is maintained because, after a half-cycle, the sign of the field reverses, as does the sign of the charge. Therefore, single-directional flow develops in the mother droplet along the needle electrode. However, some time is required to induce charge in a double layer under an AC electric field. The characteristic charging time for a double-layer bulk

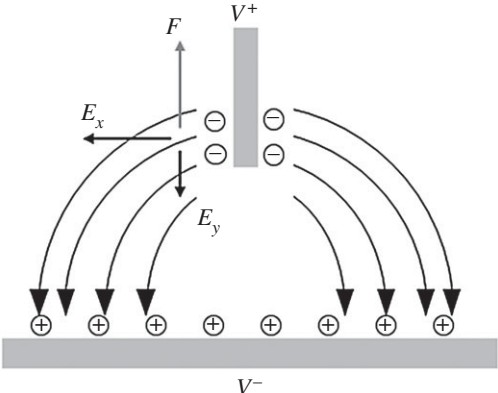

**Figure 8.** The schematic of the charge induced in response to applied potentials.

electrolyte is given by $\tau = RC$, where $R$ is the bulk electrolyte resistance and is $C$ the capacitance of the double layer. The characteristic time for a 0.1 mM KCl solution is several tens of microseconds [28,30]. Therefore, electro-osmotic flow attributable to induced charge has a frequency of less than 1 MHz [22,23,25,31]. In the present study, the flow frequency range was several tens of MHz; it is thus difficult to attribute the phenomenon to AC electro-osmosis caused by induced charge. Lastochkin *et al.* reported that AC electro-osmosis was not in fact attributable to induced charge, but rather to Faradaic polarization at a higher frequency. In such a case, ions near the electrode are of the opposite sign to that of the induced charge; the net flow is thus in the opposite direction [31].

Some studies, including Fuhr *et al.*, reported fluid motion in a microchannel under an AC electric field, referred to as 'travelling wave', based on electrode arrays with different phases [32–35]; localized heating created permittivity and conductivity gradients in the fluid, potentially giving rise to electrical volume forces (electrothermal forces). By contrast, the fluid motion was caused by a pair of electrodes in the present study; however, the travelling wave required electrode arrays with sequential phases. Additionally, Fuhr *et al.* suggested that the frequency of peak velocity was proportional to the conductivity of the solution, whereas we observed that the particular frequency of the peak velocity was similar in the range from 0.01 to 100 mM in KCl solution.

Wu *et al.* showed AC electro-osmotic flow along the electrode in an open microchannel, referred to as 'rectified AC electroosmosis' [36]; although this approach appears to have a geometry similar to our own, the results obtained in this study showed several differences. First, the fluid velocity under an AC electric field was slower than that under a DC electric field. This is the complete opposite of our results. Moreover, rectified AC electro-osmosis required a relatively low frequency (about 10 Hz) and a high (a few hundred volts) gate voltage.

## 5. Conclusion

In summary, we report the one-directional flow along fine electrodes under an AC electric field. The one-directional flow was generated at a specific voltage (above 5 $V_{PP}$) and frequency domain (approx. MHz). Several interesting results were observed. First, the flow rates with the AC electric field were much faster ($10^3$-fold higher) than that with a DC electric field. The flow rates of transported ionic solutions increased with the applied voltage and ionic concentration of the KCl solution. Second, the flow rates of the transported ionic solutions changed by altering the frequency, showing Gaussian behaviour, and there was a particular frequency that maximized the flow rates of transported ionic solutions. Importantly, one-directional flows along CNT nanowires were generated with no additional equipment or frequency matching. Considering the molar concentration of the ionic solutions, the frequency ranges for generating one-directional flow was reduced as the ionic concentration decreased from 1 to 100 mM. Additional research is needed to understand the mechanism of one-directional flow under the AC electric field, and dielectric breakdown and acoustic streaming may be a candidate of possible mechanism. We expect that the discovered phenomenon is applicable to a variety of areas, including microfluidic systems and drug injection into a single cell.

Ethics. The all experiments were conducted in the laboratory of Department of Mechanical Engineering, Pohang University of Science and Technology (POSTECH). All experimental results included in this paper were tested repeatedly and confirmed to be repeatable.

Data accessibility. We include all the experimental data in the manuscript or in the electronic supplementary material.

Authors' contributions. J.H.S., I.S.K., H.K., W.C. and G.L. designed experiments. J.H.S, K.K. and H.W. carried out the experiments, and J.H.S wrote this manuscript and created the figures. W.C. and G.L. supervised the research. All authors discussed the results and commented on the manuscript.

Competing interests. We declare we have no competing interests.

Funding. This work was supported by the National Research Foundation of Korea (NRF) grant funded by the Korea government (MEST) (no. 2015R1A2A1A14027903) and by the Korea government (MSIP) (no. NRF-2016R1C1B1015521).

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
