## [Reviewer comments · Royal Society Open Science]

Review History

RSOS-180657.R0 (Original submission)

Review form: Reviewer 1 (Brian Storey)

Is the manuscript scientifically sound in its present form?

Yes

Are the interpretations and conclusions justified by the results?

Yes

Is the language acceptable?

Yes

Is it clear how to access all supporting data?

No

Do you have any ethical concerns with this paper?

No

Have you any concerns about statistical analyses in this paper?

Yes

Recommendation?

Accept with minor revision (please list in comments)

Comments to the Author(s)

Overall the paper is interesting - it reports on some unusual experimental results that are not easily explained by any theory I am aware of. In the intro, the authors review several possibilities and conclude none of them explain. I have a few comments in no particular order.

- 1) The authors claim there are no reactions because there are no bubbles. Did they try to assess reactions in a more quantitative way, or does the small droplet size make this difficult - i.e. measure pH?
- 2) I didn't really understand the brackets and the stars in all the figures? I found it distracting from the otherwise pretty simple plots.
- 3) I didn't see a comment in the paper about how many repeats were done at each measurement. Are the error bars the scatter at that condition? Is it the $n=7$? If so, be more explicit - i.e. say each experiment was repeated 7 times. If it was repeated, was it with a new CNT or the same CNT but different flow measurement? Just a sentence or two to explain how repeatable your results are.
- 4) The table at the end could be better formatted - or maybe all this data would be organized as supplementary material. In any case it was hard to figure out what you were showing there.
- 5) I don't understand why at 40 MHz you have 200 nL/min but can't measure lower than that. A little more explanation would be helpful here. I don't doubt the difficulty due to the scale here.
- 6) Caption for figure 7 plots frequency on one axis and then reports a frequency in the caption. That seems confusing.
- 7) The comment in the caption about values compared by a t-test seems unusual. I don't really know what you mean by this comment. Are you trying to claim the data are statistically significantly different? Why not just show the data with the error bars?

Review form: Reviewer 2

Is the manuscript scientifically sound in its present form?

Yes

Are the interpretations and conclusions justified by the results?

No

Is the language acceptable?

Yes

Is it clear how to access all supporting data?

Not Applicable

Do you have any ethical concerns with this paper?

No

Have you any concerns about statistical analyses in this paper?

I do not feel qualified to assess the statistics

Recommendation?

Major revision is needed (please make suggestions in comments)

Comments to the Author(s)

The authors explore the transport of aqueous electrolytes under imposed high frequency (MHz) electric fields in a geometry that consists of a planar electrode upon which the electrolyte drop sits and a vertically held carbon nanotube coated tungsten wire in contact with the drop. Under certain conditions, they observe that the electrolyte migrate up the tungsten wire.

They make the assertion that this is a previously unreported behavior and that it cannot be explained by existing theories, namely, dielectrophoresis, induced charge electroosmosis, travelling wave, AC, and rectified AC electroosmosis. While this may indeed be a novel phenomenon, it is my opinion that they have not done an adequate job of exploring the details of the other theories that they present, nor have they included all electrokinetic phenomena. Furthermore, they present their experimental observations, but provide no theoretical explanation for the observed behavior. I would recommend publication only after major revisions to the manuscript. My specific comments are below:

1. In the introduction, dielectrophoresis, induced charge electroosmosis, travelling wave, AC, and rectified AC electroosmosis are all listed as possibly related and the authors attempt to explain why each theory is inadequate. The treatment is qualitative and sparse. I suggest the authors include a more detailed treatment of each of these theories, presenting the relevant equations that explain fluid flow under each theory and then explain why the presented experiment does not fit with theory.

2. On line 26, the statement "the mechanism of 'ICEO' is based on the movement of small particles, however, the discovered phenomenon happens without any particles" is false. ICEO can occur around any polarizable surface. The authors cite: Squires, T. M.; Bazant, M. Z. Induced-Charge Electro-Osmosis. *Journal of Fluid Mechanics* 2004, 509, 217-252. I encourage the authors to re-read section 5 that describe ICEO flow around electrically floating electrodes in microfluidic devices.

3. There are several other electrokinetic phenomena that the authors neglect to consider as possible explanations of the observed effect:

Electrowetting: The presented setup is similar to those found in the electrowetting literature. The following review may be helpful in assessing: Mugele, Frieder, and Jean-Christophe Baret. "Electrowetting: From Basics to Applications." *Journal of Physics: Condensed Matter* 17, no. 28 (2005): R705. <https://doi.org/10.1088/0953-8984/17/28/R01>.

Electrothermal flows: For example, theories presented in: García-Sánchez, Pablo, Antonio Ramos, and Frieder Mugele. "Electrothermally Driven Flows in Ac Electrowetting." *Physical Review E* 81, no. 1 (January 2010). <https://doi.org/10.1103/PhysRevE.81.015303>. González, A., A. Ramos, H. Morgan, N. G. Green, and A. Castellanos. "Electrothermal Flows Generated by

Alternating and Rotating Electric Fields in Microsystems." *Journal of Fluid Mechanics* 564 (October 2006): 415. <https://doi.org/10.1017/S0022112006001595>.

Electrohydrodynamic flows: For example, theories presented in: Esmaeeli, Asghar, and Payam Sharifi. "Transient Electrohydrodynamics of a Liquid Drop." *Physical Review E* 84, no. 3 (2011): 036308. <https://doi.org/10.1103/PhysRevE.84.036308>. Ristenpart, W. D, I. A Aksay, and D. A Saville. "Electrohydrodynamic Flow around a Colloidal Particle near an Electrode with an Oscillating Potential." *Journal of Fluid Mechanics* 575 (2007): 83-109.

4. The authors should perform a flow visualization experiment, such as micro-PIV, to look at flow patterns in the system during the experiment. This will help to determine a possible mechanism.

5. The authors provide no competing theory to explain their observations. If it isn't a well established theory, then what is the physical mechanism that is driving the flow?

6. The authors should provide more detail on the experimental setup. Specifically, details on the electronics utilized for the experiment. For instance, what is the model number of the power supply used? How much current was flowing during the experiment? What is the conductivity of the electrolytes?

With these significant revisions, I feel that the manuscript would be more impactful to the community.

Decision letter (RSOS-180657.R0)

03-Aug-2018

Dear Professor Shin,

The editors assigned to your paper ("One-directional flow of ionic solutions along fine electrodes under an alternating current electric field") have now received comments from reviewers. We would like you to revise your paper in accordance with the referee and Associate Editor suggestions which can be found below (not including confidential reports to the Editor). Please note this decision does not guarantee eventual acceptance.

Please submit a copy of your revised paper before 26-Aug-2018. Please note that the revision deadline will expire at 00.00am on this date. If we do not hear from you within this time then it will be assumed that the paper has been withdrawn. In exceptional circumstances, extensions may be possible if agreed with the Editorial Office in advance. We do not allow multiple rounds of revision so we urge you to make every effort to fully address all of the comments at this stage. If deemed necessary by the Editors, your manuscript will be sent back to one or more of the original reviewers for assessment. If the original reviewers are not available, we may invite new reviewers.

- Data accessibility

If you wish to submit your supporting data or code to Dryad (<http://datadryad.org/>), or modify your current submission to dryad, please use the following link:
<http://datadryad.org/submit?journalID=RSOS&manu=RSOS-180657>

- Competing interests

- Authors' contributions

- Acknowledgements

- Funding statement

Please note that Royal Society Open Science charge article processing charges for all new submissions that are accepted for publication. Charges will also apply to papers transferred to Royal Society Open Science from other Royal Society Publishing journals, as well as papers submitted as part of our collaboration with the Royal Society of Chemistry (<http://rsos.royalsocietypublishing.org/chemistry>). If your manuscript is newly submitted and subsequently accepted for publication, you will be asked to pay the article processing charge, unless you request a waiver and this is approved by Royal Society Publishing. You can find out more about the charges at <http://rsos.royalsocietypublishing.org/page/charges>. Should you have any queries, please contact openscience@royalsociety.org.

Kind regards,
Andrew Dunn
Senior Publishing Editor
Royal Society Open Science Editorial Office
Royal Society Open Science
openscience@royalsociety.org

on behalf of Dr Oliver Jensen (Associate Editor) and R. Kerry Rowe (Subject Editor)
openscience@royalsociety.org

Associate Editor's comments (Dr Oliver Jensen):

Reviewers have assessed your paper and have indicated that major revisions are necessary if it is to be suitable for publication. Please therefore revise your paper in line with the recommendations of the reviewers, providing a point-by-point response to each issue raised by the reviewers. Your revised study will be subject to further review.

Comments to Author:

Reviewers' Comments to Author:

Reviewer: 1

Comments to the Author(s)

Overall the paper is interesting - it reports on some unusual experimental results that are not easily explained by any theory I am aware of. In the intro, the authors review several possibilities and conclude none of them explain. I have a few comments in no particular order.

1) The authors claim there are no reactions because there are no bubbles. Did they try to assess reactions in a more quantitative way, or does the small droplet size make this difficult - i.e. measure pH?

- 2) I didn't really understand the brackets and the stars in all the figures? I found it distracting from the otherwise pretty simple plots.
- 3) I didn't see a comment in the paper about how many repeats were done at each measurement. Are the error bars the scatter at that condition? Is it the $n=7$? If so, be more explicit - i.e. say each experiment was repeated 7 times. If it was repeated, was it with a new CNT or the same CNT but different flow measurement? Just a sentence or two to explain how repeatable your results are.
- 4) The table at the end could be better formatted - or maybe all this data would be organized as supplementary material. In any case it was hard to figure out what you were showing there.
- 5) I don't understand why at 40 MHz you have 200 nL/min but can't measure lower than that. A little more explanation would be helpful here. I don't doubt the difficulty due to the scale here.
- 6) Caption for figure 7 plots frequency on one axis and then reports a frequency in the caption. That seems confusing.
- 7) The comment in the caption about values compared by a t-test seems unusual. I don't really know what you mean by this comment. Are you trying to claim the data are statistically significantly different? Why not just show the data with the error bars?

Reviewer: 2

Comments to the Author(s)

The authors explore the transport of aqueous electrolytes under imposed high frequency (MHz) electric fields in a geometry that consists of a planar electrode upon which the electrolyte drop sits and a vertically held carbon nanotube coated tungsten wire in contact with the drop. Under certain conditions, they observe that the electrolyte migrate up the tungsten wire.

They make the assertion that this is a previously unreported behavior and that it cannot be explained by existing theories, namely, dielectrophoresis, induced charge electroosmosis, travelling wave, AC, and rectified AC electroosmosis. While this may indeed be a novel phenomenon, it is my opinion that they have not done an adequate job of exploring the details of the other theories that they present, nor have they included all electrokinetic phenomena. Furthermore, they present their experimental observations, but provide no theoretical explanation for the observed behavior. I would recommend publication only after major revisions to the manuscript. My specific comments are below:

1. In the introduction, dielectrophoresis, induced charge electroosmosis, travelling wave, AC, and rectified AC electroosmosis are all listed as possibly related and the authors attempt to explain why each theory is inadequate. The treatment is qualitative and sparse. I suggest the authors include a more detailed treatment of each of these theories, presenting the relevant equations that explain fluid flow under each theory and then explain why the presented experiment does not fit with theory.
2. On line 26, the statement "the mechanism of 'ICEO' is based on the movement of small particles, however, the discovered phenomenon happens without any particles" is false. ICEO can occur around any polarizable surface. The authors cite: Squires, T. M.; Bazant, M. Z. Induced-Charge Electro-Osmosis. *Journal of Fluid Mechanics* 2004, 509, 217-252. I encourage the authors to re-read section 5 that describe ICEO flow around electrically floating electrodes in microfluidic devices.

3. There are several other electrokinetic phenomena that the authors neglect to consider as possible explanations of the observed effect:

Electrowetting: The presented setup is similar to those found in the electrowetting literature. The following review may be helpful in assessing: Mugele, Frieder, and Jean-Christophe Baret. "Electrowetting: From Basics to Applications." *Journal of Physics: Condensed Matter* 17, no. 28 (2005): R705. <https://doi.org/10.1088/0953-8984/17/28/R01>.

Electrothermal flows: For example, theories presented in: García-Sánchez, Pablo, Antonio Ramos, and Frieder Mugele. "Electrothermally Driven Flows in Ac Electrowetting." *Physical Review E* 81, no. 1 (January 2010). <https://doi.org/10.1103/PhysRevE.81.015303>. González, A., A. Ramos, H. Morgan, N. G. Green, and A. Castellanos. "Electrothermal Flows Generated by Alternating and Rotating Electric Fields in Microsystems." *Journal of Fluid Mechanics* 564 (October 2006): 415. <https://doi.org/10.1017/S0022112006001595>.

Electrohydrodynamic flows: For example, theories presented in: Esmaeeli, Asghar, and Payam Sharifi. "Transient Electrohydrodynamics of a Liquid Drop." *Physical Review E* 84, no. 3 (2011): 036308. <https://doi.org/10.1103/PhysRevE.84.036308>. Ristenpart, W. D, I. A Aksay, and D. A Saville. "Electrohydrodynamic Flow around a Colloidal Particle near an Electrode with an Oscillating Potential." *Journal of Fluid Mechanics* 575 (2007): 83-109.

4. The authors should perform a flow visualization experiment, such as micro-PIV, to look at flow patterns in the system during the experiment. This will help to determine a possible mechanism.

5. The authors provide no competing theory to explain their observations. If it isn't a well established theory, then what is the physical mechanism that is driving the flow?

6. The authors should provide more detail on the experimental setup. Specifically, details on the electronics utilized for the experiment. For instance, what is the model number of the power supply used? How much current was flowing during the experiment? What is the conductivity of the electrolytes?

With these significant revisions, I feel that the manuscript would be more impactful to the community.

Author's Response to Decision Letter for (RSOS-180657.R0)

Thank you for your kind handling of our manuscript. In this manuscript, we responded to the reviewer comments and all changes in our manuscript were highlighted.

RSOS-180657.R1 (Revision)

Review form: Reviewer 2

Is the manuscript scientifically sound in its present form?

Yes

Are the interpretations and conclusions justified by the results?

No

Is the language acceptable?

Yes

Is it clear how to access all supporting data?

Yes

Do you have any ethical concerns with this paper?

No

Have you any concerns about statistical analyses in this paper?

I do not feel qualified to assess the statistics

Recommendation?

Accept with minor revision (please list in comments)

Comments to the Author(s)

The authors have included additional details of electrokinetic theories in the text as I recommended in the initial review; however, some of the analysis given, in particular, on electrowetting is difficult to follow. I believe more analysis needs to be performed beyond the qualitative explanation given by the authors. In specific:

Will the authors please provide more explanation for this statement: "However, electrowetting differs from the phenomenon we describe in that an insulating layer is placed between the mother droplet and the plane electrode; this renders monodirectional flow impossible"

To my knowledge, electrowetting usually performed on a dielectric layer to limit Faradaic reactions. Electrowetting can and does occur on bare metal electrodes as well. Regardless, I do not see how the presence of an insulator means that monodirectional flow is impossible. Please explain.

Also, the field is very high near the tip of the nanotube. The authors state that the CNT has a diameter of ~ 1 nm, and the effect appears at >5 Vpp. This means that peak electric fields near the CNT tip are on the order of 10^9 V/m. This is much higher than the dielectric breakdown strength of water which is $\sim 70 \times 10^6$ V/m. Is it possible that this effect is due to electrical breakdown?

Decision letter (RSOS-180657.R1)

11-Dec-2018

Dear Professor Shin:

On behalf of the Editors, I am pleased to inform you that your Manuscript RSOS-180657.R1 entitled "One-directional flow of ionic solutions along fine electrodes under an alternating current electric field" has been accepted for publication in Royal Society Open Science subject to minor revision in accordance with the referee suggestions. Please find the referees' comments at the end of this email.

The reviewers and Subject Editor have recommended publication, but also suggest some minor revisions to your manuscript. Therefore, I invite you to respond to the comments and revise your manuscript.

- Ethics statement

- Data accessibility

<http://datadryad.org/submit?journalID=RSOS&manu=RSOS-180657.R1>

- Competing interests

- Authors' contributions

- Acknowledgements

- Funding statement

Because the schedule for publication is very tight, it is a condition of publication that you submit the revised version of your manuscript before 20-Dec-2018. Please note that the revision deadline will expire at 00.00am on this date. If you do not think you will be able to meet this date please let me know immediately.

Kind regards,
Andrew Dunn

Royal Society Open Science Editorial Office
Royal Society Open Science
openscience@royalsociety.org

on behalf of Dr Oliver Jensen (Associate Editor) and R. Kerry Rowe (Subject Editor)
openscience@royalsociety.org

Associate Editor Comments to Author (Dr Oliver Jensen):

Associate Editor: 1

Comments to the Author:

Please make a further revision of your paper to address the specific points raised by the reviewer. In particular, please remove the assertion (sentence 2 of the Conclusions) that "The discovered phenomenon ... cannot be explained using previous theories." This comment is not fully supported by your analysis.

In addition to dielectric breakdown as a candidate mechanism of the effect, please also comment on the possibility of acoustic (steady) streaming as a possible mechanism.

Reviewer comments to Author:

Reviewer: 2

Comments to the Author(s)

The authors have included additional details of electrokinetic theories in the text as I recommended in the initial review; however, some of the analysis given, in particular, on electrowetting is difficult to follow. I believe more analysis needs to be performed beyond the qualitative explanation given by the authors. In specific:

Will the authors please provide more explanation for this statement: "However, electrowetting differs from the phenomenon we describe in that an insulating layer is placed between the mother droplet and the plane electrode; this renders monodirectional flow impossible"

To my knowledge, electrowetting usually performed on a dielectric layer to limit Faradaic reactions. Electrowetting can and does occur on bare metal electrodes as well. Regardless, I do not see how the presence of an insulator means that monodirectional flow is impossible. Please explain.

Also, the field is very high near the tip of the nanotube. The authors state that the CNT has a diameter of ~1 nm, and the effect appears at >5Vpp. This means that peak electric fields near the CNT tip are on the order of 10^9 V/m. This is much higher than the dielectric breakdown strength of water which is $\sim 70 \times 10^6$ V/m. Is it possible that this effect is due to electrical breakdown?

Author's Response to Decision Letter for (RSOS-180657.R1)

See Appendix A.

Decision letter (RSOS-180657.R2)

11-Jan-2019

Dear Professor Shin,

I am pleased to inform you that your manuscript entitled "One-directional flow of ionic solutions along fine electrodes under an alternating current electric field" is now accepted for publication in Royal Society Open Science.

on behalf of Dr Oliver Jensen (Associate Editor) and R. Kerry Rowe (Subject Editor)
openscience@royalsociety.org

Appendix A

Associate Editor Comments to Author (Dr Oliver Jensen):

Associate Editor: 1

Comments to the Author:

Please make a further revision of your paper to address the specific points raised by the reviewer. In particular, please remove the assertion (sentence 2 of the Conclusions) that "The discovered phenomenon ... cannot be explained using previous theories." This comment is not fully supported by your analysis.

In addition to dielectric breakdown as a candidate mechanism of the effect, please also comment on the possibility of acoustic (steady) streaming as a possible mechanism.

: We changed the manuscript considering your comments.

Acoustic streaming is a steady flow caused by the absorption of high amplitude of acoustic vibration. According to Ko et al., the amplitude of the liquid air interface decreases as the frequency at which the AC electric field applied, and vibration is hardly observed at frequencies higher than 8 kHz [1]. Since AC electric field with frequencies of several tens of MHz is applied in this experiment, it is difficult to deduce it as flow by acoustic streaming.

[1] Ko et al., "Hydrodynamic Flows in Electrowetting", Langmuir, 2008, 24, 1094

Reviewer comments to Author:

Reviewer: 2

Comments to the Author(s)

The authors have included additional details of electrokinetic theories in the text as I recommended in the initial review; however, some of the analysis given, in particular, on electrowetting is difficult to follow. I believe more analysis needs to be performed beyond the qualitative explanation given by the authors. In specific:

Will the authors please provide more explanation for this statement:

"However, electrowetting differs from the phenomenon we describe in that an insulating layer is placed between the mother droplet and the plane electrode; this renders monodirectional flow impossible"

To my knowledge, electrowetting usually performed on a dielectric layer to limit Faradaic reactions. Electrowetting can and does occur on bare metal electrodes a well. Regardless, I do not see how the presences of an insulator mean that monodirectional flow is impossible. Please explain.

: We looked up the relevant data after the reviewer's opinion that electrowetting can and does occur on bare metal electrodes a well. In our investigation, electrowetting is generally difficult to occur on the surface of solid conductor (1) and in the case of liquid metals such as Ga, electrowetting can occur on metal surfaces because a thin oxide layer is formed on the metal surface (2). We have found a paper on electrowetting on bare electrodes using stainless steel as an electrode (3). In this case, the response was much slower than EWOD. One directional liquid pumping occurs at frequencies of tens of MHz and it is higher than ordinary AC EWOD frequencies. Since the response time of electrowetting on bare metal electrode is much slower than that of EWOD, which will lead to a larger difference in the frequency range. In addition, liquid pumping does not be observed when as insulating layer laid on the gold electrode surface such as EWOD structures.

(1) Deborah J. Lomax et al. "Ultra-low voltage electrowetting using graphite surface", *Soft Matter*, 2016, 12, 8798

(2) Collin B. Eaker et al. "Electrowetting without external voltage using paint-on electrodes", *Lab Chip*, 2017, 17, 1069

(3) Yanna Liu et al. "Ultralow voltage irreversible electrowetting dynamics of an aqueous drop on a stainless steel surface" Langmuir, 2015,31,3840

Also, the field is very high near the tip of the nanotube. The authors state that the CNT has a diameter of ~ 1 nm, and the effect appears at $>5V_{pp}$. This means that peak electric fields near the CNT tip are on the order of 10^9 V/m. This is much higher than the dielectric breakdown strength of water which is $\sim 70 \times 10^6$ V/m. Is it possible that this effect is due to electrical breakdown?

: The diameter of CNT nanowires coated by gold nanoparticles were about 800 nm. Thus, the peak electric fields near CNT nanowires are on the order of ~ 6.25 M V/m. This value is smaller than the dielectric breakdown strength of water.